Spatial distribution of environmental DNA in a nearshore marine habitat

O’Donnell James L. jimmyod@uw.edu 1
Kelly Ryan P. 1
Shelton Andrew Olaf 2
Samhouri Jameal F. 3
Lowell Natalie C. 1 4
Williams Gregory D. 5
1 School of Marine and Environmental Affairs, University of Washington , Seattle , WA , United States of America
2 Earth Resource Technology, Inc., Under Contract to the Northwest Fisheries Science Center, National Marine Fisheries Service, National Oceanic and Atmospheric Administration , Seattle , WA , United States of America
3 Conservation Biology Division, Northwest Fisheries Science Center, National Marine Fisheries Service, National Oceanic and Atmospheric Administration , Seattle , WA , United States of America
4 School of Aquatic and Fishery Sciences, University of Washington , Seattle , WA , United States of America
5 Pacific States Marine Fisheries Commission, Under Contract to the Northwest Fisheries Science Center, National Marine Fisheries Service, National Oceanic and Atmospheric Administration , Seattle , WA , United States of America
Johnson Magnus
Electronic publication date: 2017 Feb 28
Publication date: 2017
Volume: 5
Electronic Location ID: e3044
Received 2016 Nov 28; Accepted 2017 Jan 29
Copyright year: 2017
License: This is an open access article, free of all copyright, made available under the Creative Commons Public Domain Dedication. This work may be freely reproduced, distributed, transmitted, modified, built upon, or otherwise used by anyone for any lawful purpose.
License URL: https://creativecommons.org/publicdomain/zero/1.0/

Keywords: Marine, Metabarcoding, Metagenomics, Estuarine, Molecular ecology, Environmental monitoring

Funding: David and Lucile Packard Foundation 2014-39827 This work was supported by a grant from the David and Lucile Packard Foundation to RPK (grant 2014-39827). The funders had no role in study design, data collection and analysis, decision to publish, or preparation of the manuscript.

==============================
In the face of increasing threats to biodiversity, the advancement of methods for surveying biological communities is a major priority for ecologists. Recent advances in molecular biological technologies have made it possible to detect and sequence DNA from environmental samples (environmental DNA or eDNA); however, eDNA techniques have not yet seen widespread adoption as a routine method for biological surveillance primarily due to gaps in our understanding of the dynamics of eDNA in space and time. In order to identify the effective spatial scale of this approach in a dynamic marine environment, we collected marine surface water samples from transects ranging from the intertidal zone to four kilometers from shore. Using PCR primers that target a diverse assemblage of metazoans, we amplified a region of mitochondrial 16S rDNA from the samples and sequenced the products on an Illumina platform in order to detect communities and quantify their spatial patterns using a variety of statistical tools. We find evidence for multiple, discrete eDNA communities in this habitat, and show that these communities decrease in similarity as they become further apart. Offshore communities tend to be richer but less even than those inshore, though diversity was not spatially autocorrelated. Taxon-specific relative abundance coincided with our expectations of spatial distribution in taxa lacking a microscopic, pelagic life-history stage, though most of the taxa detected do not meet these criteria. Finally, we use carefully replicated laboratory procedures to show that laboratory treatments were remarkably similar in most cases, while allowing us to detect a faulty replicate, emphasizing the importance of replication to metabarcoding studies. While there is much work to be done before eDNA techniques can be confidently deployed as a standard method for ecological monitoring, this study serves as a first analysis of diversity at the fine spatial scales relevant to marine ecologists and confirms the promise of eDNA in dynamic environments.

Introduction

The patterns and causes of variability in ecological communities across space are both seminal and contentious areas of study in ecology (Hubbell, 2001; Anderson et al., 2011). One consistently observed pattern of community spatial heterogeneity is that communities close to one another tend to be more similar than those that are farther apart (Nekola & White, 1999). This decrease in community similarity with increasing spatial separation is called distance decay and has been reported from communities of tropical trees (Condit, 2002; Chust et al., 2006), ectomycorrhizal fungi (Bahram et al., 2013), salt marsh plants (Guo et al., 2015), and microorganisms (Martiny et al., 2011; Chust et al., 2013; Wetzel et al., 2012; Bell, 2010). Typically, this relationship is assessed by regressing a measure of community similarity against a measure of spatial separation for a set of sites at which a set of species’ abundances (or presences) is calculated. Yet no existing biodiversity survey method completely censuses all of the organisms in a given area. The lack of a single ‘silver bullet’ method of sampling contributes inconclusiveness to the study of spatial patterning in ecology (Levin, 1992), and leaves open the possibility of new and more comprehensive methods.

From a boat or aircraft, scientists can count whales by sight, but not the krill on which they feed. For example, towed fishing nets can efficiently sample organisms larger than the mesh and slower than the boat, but overlook viruses and have undesirable effects on charismatic air-breathing species. However, DNA-based surveys show great promise as an efficient technique for detecting a previously unthinkable breadth of organisms from a single sample.

Microbiologists have used nucleic acid sequencing to quantify the composition and function of microbial communities in a wide variety of habitats (Handelsman et al., 1998; Tyson et al., 2004; Venter et al., 2004; Iverson et al., 2012). To do so, microorganisms are collected in a sample of environmental medium (e.g., water), their DNA or RNA is isolated and sequenced, and the identity and abundance of sequences is considered to reflect the community of organisms contained in the sample, which indirectly estimates the quantity of organisms in an area.

Macroorganisms shed DNA-containing cells into the environment (environmental DNA or eDNA) that can be sampled in the same way (Ficetola et al., 2008; Thomsen et al., 2012). Potentially, eDNA methods allow a broad swath of macroorganisms to be surveyed from basic environmental samples. However, the accuracy and reliability of indirect estimates of macroorganismal abundance has been debated because the entire organisms are not contained within the sample (Cowart et al., 2015). Concern surrounding eDNA methods is rooted in uncertainty about the attributes of eDNA in the environment relative to actual organisms (Shelton et al., 2016; Evans et al., 2016). Basic questions such as how long DNA can persist in that environment and how far DNA can travel remain largely unknown (but see Klymus et al. (2015); Turner, Uy & Everhart (2015); Strickler, Fremier & Goldberg (2015); Deiner & Altermatt (2014)) and impede inference about local organismal presence from an environmental sample. As a result, estimating the spatial and temporal resolution of eDNA studies in the field is a key step in making these methods practical.

The relationship between local organismal abundance and eDNA is further complicated in habitats where the environmental medium itself may transport eDNA away from its source. We know that genetic material can move away from its source precisely because organisms can be detected indirectly without being present in the sample (Kelly et al., 2016b). One might reasonably expect eDNA to travel farther in a highly dynamic fluid such as the open ocean or flowing river than it would through the sediment at the bottom of a stagnant pond (Deiner & Altermatt, 2014; Shogren et al., 2016). Yet even studies of extremely dynamic habitats such as coastlines with high wave energy have found remarkable evidence that eDNA transport is limited enough that DNA methods can detect differences among communities separated by less than 100 m (Port et al., 2016).

While rigorous laboratory studies have investigated the effects of some environmental factors on eDNA persistence (Klymus et al., 2015; Barnes et al., 2014; Sassoubre et al., 2016) and the transport of eDNA in specific contexts (Deiner & Altermatt, 2014), we suggest that field studies comparing the spatial distribution of communities of eDNA with expectations based on prior knowledge of organisms’ distributions are also critical to developing a working understanding of eDNA in the real world. Research to date has documented the non-random spatial distribution of meiofaunal (Fonseca et al., 2014; Guardiola et al., 2016), microbial (Lallias et al., 2015), and extracellular (Guardiola et al., 2015) eDNA of marine and estuarine sediments, and of microscopic plankton in open ocean waters (De Vargas et al., 2015). These studies conducted targeted sampling at intermediate (thousands of meters) to global (thousands of kilometers) scales. Here, we use a grid-based environmental sampling strategy to assess spatial variability of eDNA in a coastal marine environment at a fine scale (tens to thousands of meters), using molecular methods that focus on macrobial metazoans.

We apply methods derived from community ecology to understand spatial patterns and patchiness of eDNA. The underlying mechanism thought to drive the slope of the distance decay relationship in ecological communities is the rate of movement of individuals among sites, which may be driven by underlying processes such as habitat suitability. Because eDNA is shed and transported away from its source, the increased movement of eDNA particles should homogenize community similarity, and thus erode the distance decay relationship of eDNA communities.

Puget Sound is a deep, narrow fjord in Washington, USA, where a narrow band of shallow bottom hugs the shoreline and abruptly gives way to a central depth of up to 300 m. This form allows the juxtaposition of communities associated with distinctly different habitats: shallow, intertidal benthos, and euphotic pelagic (Burns, 1985). At the upper reaches of the intertidal, the shoreline substrate varies from soft, fine sediment to cobble and boulder rubble. Soft intertidal sediments are inhabited by burrowing bivalves (Bivalvia), segmented worms (Annelida), and acorn worms (Enteropneusta), and in some lower intertidal and high subtidal ranges by eelgrass (Zostera marina) (Kozloff, 1973; Dethier, 2010). Eelgrass meadows harbor epifaunal and infaunal biota, and attract transient species which use the meadows for shelter and to feed on resident organisms. Hard intertidal surfaces support a well-documented biota including barnacles (Sessilia) and other crustaceans, mussels (Bivalvia:Mytilidae), anemones (Actiniaria), sea stars (Asteroidea), urchins (Echinoidea), Bryozoans (Ectoprocta), crustaceans (Decapoda), and a variety of algae (Dethier, 2010). Hard bottoms of the lower intertidal and high subtidal are home to macroalgae such as Laminariales and Desmarestiales which provide habitat for a distinct community of fish and invertebrates. The upper pelagic is home to a diverse assemblage of microscopic plankton including diatoms, copepods, and larvae (Strickland, 1983), as well as transitory fish and marine mammals.

We took advantage of this setting to explore the spatial variation and distribution of marine eDNA communities. Using PCR-based methods and massively parallel sequencing, we surveyed mitochondrial 16S sequences from a suite of marine animals in water samples collected over a grid of sites extending from the shoreline out to 4 kilometer offshore in Puget Sound, Washington, USA. We leverage this sampling design to perform an explicitly spatial analysis of eDNA-derived community similarity. We investigate two primary objectives. First we examine the spatial patterning of eDNA and determine the degree to which eDNA community similarity can be predicted by physical proximity. We expect that physical proximity will be a strong predictor of community similarity, and that community differences can be detected over small distances. Second, we examine the distribution of diversity from eDNA data, and compare it to our expectations based on distributions of macrobial communities. We expect that distinct eDNA communities exist in this setting, and that their spatial distribution coincides with that of adult macrobial organisms. Because of the vastly different communities of benthic macrobial metazoans as a function of distance from shore, we expect that more than one eDNA community is present across our 4 kilometer sampling grid, and that communities change as a function of distance from shore. For this reason, we examine two diversity measures of eDNA communities that have been widely used to reveal broad scale patterns based on macrobiota in many ecological systems. Finally, we identify the taxa represented in the eDNA communities, which span a range of life-history characteristics, and we expect that the spatial distribution of eDNA will most closely resemble the distribution of adults in taxa with low dispersal potential.

Methods

There are seven discrete steps to our methodology: (1) Environmental sample collection, (2) isolation of particulates from water via filtration, (3) isolation of DNA from filter membrane, (4) amplification of target locus via PCR, (5) sequencing of amplicons, (6) bioinformatic translation of raw sequence data into tables of sequence abundance among samples, and (7) community ecological analyses of eDNA. We provide brief overviews of these steps here, and encourage the reader to review the fully detailed methods presented in Supplemental Information 1.

Environmental sampling

Starting from lower-intertidal patches of Zostera marina, we collected water samples at 1 m depth from 8 points (0, 75, 125, 250, 500, 1,000, 2,000, and 4,000 m) along three parallel transects separated by 1,000 m (24 sample locations total; Fig. 1). Samples were collected by attaching bottles to a PVC pole and lowering it over the side of a boat over the span of one hour on 27 June 2014. To destroy residual DNA on equipment used for field sampling and filtration, we washed with a 1:10 solution of household bleach (8.25% sodium hypochlorite; 7.25% available chlorine) and deionized water, followed by thorough rinsing with deionized water. Each environmental sample was collected in a clean 1 liter high-density polyethylene bottle, the opening of which was covered with 500 micrometer nylon mesh to prevent entry of larger particles. Immediately after collecting the sample, the mesh was replaced with a clean lid and the sample was held on ice until filtering.

Figure 1 Map of study area.

Depth in meters below sea level is indicated by shading and 25 meter contours. Sampled locations are indicated by red points.

Filtration

One liter from each water sample was filtered in the lab on a clean polysulfone vacuum filter holder fitted with a 47 millimeter diameter cellulose acetate membrane with 0.45 micrometer pores. Filter membranes were moved into 900 microliters of Longmire buffer (Longmire, Maltbie & Baker, 1997) using clean forceps and stored at room temperature (Renshaw et al., 2015). To test for the extent of contamination attributable to laboratory procedures, we filtered three replicate 1 liter samples of deionized water. These samples were treated identically to the environmental samples throughout the remaining protocols.

DNA purification

DNA was purified from the membrane following a phenol:chloroform:isoamyl alcohol protocol following Renshaw (Renshaw et al., 2015). Preserved membranes were incubated at 65 °C for 30 min before adding 900 microliters of phenol:chloroform:isoamyl alcohol and shaking vigorously for 60 s. We conducted two consecutive chloroform washes by centrifuging at 14,000 rpm for 5 min, transferring the aqueous layer to 700 microliters chloroform, and shaking vigorously for 60 s. After a third centrifugation, 500 microliters of the aqueous layer was transferred to tubes containing 20 microliters 5 molar NaCl and 500 microliters 100% isopropanol, and frozen at −20 °C for approximately 15 h. Finally, all liquid was removed by centrifuging at 14,000 rpm for 10 min, pouring off or pipetting out any remaining liquid, and drying in a vacuum centrifuge at 45 °C for 15 min. DNA was resuspended in 200 microliters of ultrapure water. Four replicates of genomic DNA extracted from tissue of a species absent from the sampled environment (Oreochromis niloticus) served as positive control for the remaining protocols.

PCR amplification

We chose a primer set that amplifies an approximately 115 base pair (bp) region of the mitochondrial 16S rRNA gene in at least 10 metazoan phyla from this habitat, excludes non-metazoans, and resolves taxonomy to the family level in most cases using a public sequence database (Kelly et al., 2016a). We used a two-step polymerase chain reaction (PCR) protocol described by O’Donnell et al. (2016) to generate four replicate products from each DNA sample. In the first set of reactions, primers were identical in every reaction (forward: AGTTACYYTAGGGATAACAGCG; reverse: CCGGTCTGAACTCAGATCAYGT); primers in the second set of reactions included these same sequences but with 3 variable nucleotides (NNN) and an index sequence on the 5′ end (see Sequencing Metadata). We used the program OligoTag (Coissac, 2012) to generate 30 unique 6-nucleotide index sequences differing by a minimum Hamming distance of 3 (see Sequencing Metadata). Indexed primers were assigned to samples randomly, with the identical index sequence on the forward and reverse primer to avoid errors associated with dual-indexed multiplexing (Schnell, Bohmann & Gilbert, 2015). In a UV-sterilized hood, we prepared 25 microliter reactions containing 18.375 microliters ultrapure water, 2.5 microliters 10× buffer, 0.625 microliters deoxynucleotide solution (8 millimolar), 1 microliter each forward and reverse primer (10 micromolar, obtained lyophilized from Integrated DNA Technologies; Coralville, IA, USA), 0.25 microliters Qiagen HotStar Taq polymerase, and 1.25 microliter genomic or eDNA template at 1:100 dilution in ultrapure water. PCR thermal profiles began with an initialization step (95 °C; 15 min) followed by cycles (40 and 20 for the first and second reaction, respectively) of denaturation (95 °C; 15 s), annealing (61 °C; 30 s), and extension (72 °C; 30 s). A total of 20 identical PCRs were conducted from each DNA extract using non-indexed primers; these were pooled into four groups of five in order to ensure ample template for the subsequent PCR with indexed primers. In order to isolate the fragment of interest from primer dimer and other spurious fragments generated in the first PCR, we used the AxyPrep Mag FragmentSelect-I kit with solid-phase reversible immobilization (SPRI) paramagnetic beads at 2.5× the volume of PCR product (Axygen BioSciences, Corning, NY, USA). A 1:5 dilution in ultrapure water of the product was used as template for the second reaction. PCR products of the second reaction were purified using the Qiagen MinElute PCR Purification Kit (Qiagen, Hilden, Germany). Ultrapure water was used in place of template DNA and run along with each batch of PCRs to serve as a negative control for PCR; none of these produced visible bands on an agarose gel. In total, four separate replicates from each of 31 DNA samples were carried through the two-step PCR process for a total of 124 sequenced PCR products. These were combined with additional samples from other projects, totaling 345 samples for sequencing.

DNA sequencing

Up to 30 PCR products were combined according to their primer index in equal concentration into one of 14 pools, and 150 nanograms from each were prepared for library sequencing using the KAPA high-throughput library prep kit with real-time library amplification protocol (KAPA Biosystems, Wilmington, MA, USA). Each of these ligated sequencing adapters included an additional six base pair index sequence (NEXTflex DNA barcodes; BIOO Scientific, Austin, TX, USA). Thus, each PCR product was identifiable via its unique combination of index sequences in the sequencing adapters and primers. Fragment size distribution and concentration of each library was quantified using an Agilent 2100 BioAnalyzer. Libraries were pooled in equal concentrations and sequenced for 150 base pairs in both directions (PE150) using an Illumina NextSeq at the Stanford Functional Genomics Facility, where 20% PhiX Control v3 was added to act as a sequencing control and to enhance sequencing depth by increasing sequence diversity. Raw sequence data in fastq format is publicly available (see Data Availability).

Sequence Data Processing (Bioinformatics)

Detailed bioinformatic methods are provided in the supplemental material, and analysis scripts used from raw sequencer output onward can be found in the public project directory (see Analysis Scripts). Briefly, we performed five steps to process the sequence data: (1) Merge paired-end reads (Zhang et al., 2014), (2) eliminate low-quality reads (Edgar, 2010; Rognes et al., 2016), (3) eliminate PCR artifacts (chimeras) (Edgar, 2010; Rognes et al., 2016; Martin, 2011), (4) cluster reads by similarity into operational taxonomic units (OTUs) (Mahé et al., 2014), and (5) match observed sequences to taxon names (Camacho et al., 2009; Chamberlain & Szöcs, 2013; Chamberlain et al., 2016). Additionally, we checked for consistency among PCR replicates, excluded extremely rare sequences, and rescaled (rarefied) the data to account for differences in sequencing depth. The data for input to further analyses are a contingency table of the mean count of unique sequences, OTUs, or taxa present in each environmental sample.

Ecological analyses

After gathering the data, we use the eDNA community observed at each location to make inferences about the spatial patterning of eDNA communities. We use statistical tools from community ecology to assess the spatial structure of eDNA communities. We report similarity (1- dissimilarity) rather than dissimilarity in all cases for ease of interpretation.

Objective 1: community similarity as a function of distance

Distance decay

To address our first objective and determine whether or not nearby samples are more similar than distant ones, we fit a nonlinear model to represent decreasing community similarity with distance. We calculated the pairwise Bray–Curtis similarity (1—Bray–Curtis dissimilarity) between eDNA communities using the R package vegan (Oksanen et al., 2016) and the great circle distance between sampling points using the Haversine method as implemented by the R package geosphere (Hijmans, 2016). This model is similar to the Michaelis–Menten function, but with an asymptote fixed at 0: (1) yij=ABB+xij

Where the relationship between community similarity (yij) and spatial distance (xij) between observations i and j is determined by the similarity of samples at distance 0 (A), and the distance at which half the total change in similarity is achieved (B). This allows for samples collected very close together (near 0) to have similarity significantly less than one. We assessed model fit using the R function nls (R Core Team, 2016), using the nl2sol algorithm from the Port library to solve separable nonlinear least squares using analytically computed derivatives (http://netlib.org/port/nsg.f). We set bounds of 0 and 1 for the intercept parameter and a lower bound of 0 for the distance at half similarity; starting values of these parameters were 0.5 and xmax∕2, respectively. We calculated a 95% confidence interval for the parameters and the predicted values using a first-order Taylor expansion approach implemented by the function predictNLS in the R package propagate (Spiess, 2014).

There are other conceptually reasonable forms to expect the space-by-similarlity relationship to take; we present these in the supplemental material along with alternative data subsets and similarity indices.

Objective 2: spatial distribution of diversity

Community classification

To determine the spatial distribution and variation of eDNA communities (objective 2), we used multivariate classification algorithms. We simultaneously assessed the existence of distinct community types and the membership of samples to those community types using an unsupervised classification algorithm known as partitioning around medoids (PAM; sometimes referred to as k-medoids clustering) (Kaufman & Rousseeuw, 1990), as implemented in the R package cluster (Maechler et al., 2016). The classification of samples to communities was made on the basis of their pairwise Bray–Curtis similarity, calculated using the function vegdist in the R package vegan (Oksanen et al., 2016). Other distance metrics were evaluated but had no appreciable effect on the outcome of the analysis (Fig. S1). In order to chose an optimal number of clusters (K), we evaluated the distribution of silhouette widths, a measure of the similarity between each sample and its cluster compared to its similarity to other clusters. We repeated the analysis using fuzzy clustering (FANNY, (Kaufman & Rousseeuw, 1990)); however, the results were qualitatively similar to the results using PAM so we omit them here.

Aggregate measures of diversity

We calculated two measures of diversity, richness and evenness, to ask if aggregate metrics of the eDNA community showed evidence of spatial patterning. Richness is a measure of the number of distinct types of organisms present and so ranges from 1 (only one taxon observed) to S, the number of taxa observed across all samples. To calculate the evenness of the distribution of abundance of taxa in a sample, we used the complement of the Simpson (1949) index (1−Σpi2, where pi is the proportional abundance of taxon i). The values of this index ranges from 0 to 1, with the value interpreted as the probability that two sequences randomly selected from the sample will belong to different taxa; thus, larger values of the index indicate more evenly divided communities (Magurran, 2004). We calculated Moran’s I for both diversity metrics to test for spatial autocorrelation. We also tested for a linear effect of log-transformed distance from shore on each measure of diversity to ask how diversity changes over this strong environmental gradient.

Taxon and life history patterns

After assigning taxon names to the abundance data, we plotted the distribution in space of a selection of taxa to compare with our expectations on the basis of adult distributions (objective 2). Our aim was to understand where each taxon occurred in the greatest proportional abundance, and its distribution in space relative to that maximum. Thus, we rescaled each sample to proportional abundance, extracted the data from a single taxon, and scaled those values between 0 and 1. We collated life history characteristics for each of the major taxonomic groups recovered, including dispersal range of the gametes, larvae, and adults, adult habitat type and selectivity, and adult body size. For each life history stage of each taxon group, we made an order-of-magnitude approximation of the scale of dispersal. For example, internally fertilized species were assigned a gamete range of 0 km, while broadcast spawners were assigned a gamete range of 10 km. Similarly, adult range size was approximated as 0 km (sessile), 1 km (motile but not pelagic), or 10 km (highly mobile, pelagic). Variables were specified as ’multiple’ for life history stages known to span more than 1 magnitude of range size. For groups to which sequences were annotated with high confidence, but for which life history strategy is diverse or poorly known (e.g., families in the phylum Nemertea), we used conservative, coarse approximations at a higher taxonomic rank (see Life History Data). These data were used to contextualize group-specific spatial distributions and inform expectations based on known adult distributions.

Results

Sequence data processing (bioinformatics)

Preliminary sequence analysis strongly suggested that the observed variation among environmental samples reflects true variation in the environment, rather than variability due to lab protocols, for the following reasons (note that all value ranges are reported as mean plus and minus one standard deviation). First, all libraries passed the FastQC per-base sequence quality filter, generating a total of 371,576,190 reads passing filter generated in each direction. Second, samples in this study were represented by an adequate number of reads (333,537.9 ± 112,200.5), with no individual sample receiving fewer than 130,402 reads. Third, there was a very low frequency of cross-contamination from other libraries into those reported here (5e–005 ± 8e–05; max proportion 0.00034). Fourth, after scaling all samples to the same sequencing depth, OTUs with abundance greater than 178 reads (0.14% of a sample’s reads) experienced no turnover among PCR replicates within a sample. Fifth, sequence abundances among PCR replicates within water samples were remarkably consistent. A single sample had low similarity among PCR replicates (0.659) after removing this outlier, the lowest mean similarity among replicates within a sample was 0.966. Overall similarities among PCR replicates within a sample were extremely high (0.976 ± 0.013), and far higher than those among samples (0.3 ± 0.16). Across PCR replicates, each sample was represented by at least 781425 reads in the raw data and contained between 111 and 443 rarefied OTUs (Fig. S2).

Ecological analyses

Distance decay

Physical proximity is a good predictor of eDNA community similarity: similarity decreased from 0.40 (95% CI [0.36–0.45]) to half that amount at 4,500 m (95% CI [2,900–7,500]) (Fig. 2).

Figure 2 Distance decay relationship of environmental DNA communities.

Each point represents the Bray–Curtis similarity of a site sampled along three parallel transects comprising a 3,000 by 4,000 m grid. Blue dashed line represents fit of a nonlinear least squares regression (see ‘Methods’), and shading denotes the 95% confidence interval. Boxplot is comparisons within-sample across PCR replicates, separated by a vertical line at zero, where the central line is the median, the box encompasses the interquartile range, and the lines extend to 1.5 times the interquartile range. Boxplot outliers are omitted for clarity.

Community classification

Despite a clear trend in community similarity as a function of spatial separation, the results from our classification analysis are difficult to interpret. The silhouette analysis indicated the presence of 8 distinct communities; however, the gain in mean silhouette width from 2 was small (0.1), and lacked a distinctive peak (Fig. 3), indicating substantial uncertainty in the clustering algorithm. Thus, we present the results of cluster assignment for both K = 2 and K = 8 to illustrate the range of results (Fig. 4). Excluding taxa which occur in only one site had no discernible effect on the outcome of the PAM analysis (number of clusters, assignment to clusters). While there was no distinct spatial divide indicating the presence of an inshore versus an offshore community, one of the two communities (at K = 2) occurred in only two out of 18 samples inside 1,000 m from shore, and never occurred within 125 m of shore, suggesting the presence of an inshore and offshore community.

Figure 3 Silhouette widths from PAM analysis.

Points are the width of the PAM silhouette of each sample at each number of clusters (K). Red line is the mean, blue line is the median. Boxes encompass the interquartile range with a line at the median, and the whiskers extend to 1.5 times the interquartile range. Boxplot outliers are omitted for clarity.

Figure 4 Cluster membership of sampled sites.

Distance from onshore starting point is log scaled. Sites are colored and labeled by their assignment to a cluster by PAM analysis for number of clusters (K) chosen based on a priori expectations (A; 2) and mean silhouette width (B; 8).

Diversity in space

Sites offshore tend to be less rich and more even than those inshore (Fig. 5). Mean OTU richness declined by 1.42 per 1,000 m from a mean of 17.6 taxa (95% CI = 2.15) inshore to 11.9 taxa (95% CI = 4.31) at offshore locations (p = 0.0415; Fig. 5). Evenness increased by .0666 per 1,000 m from 0.225 (95% CI = 0.0558) to 0.491 (95% CI =  ±0.112), indicating that sequence reads were less evenly distributed among taxa in offshore samples (p ≪ 0.05; Fig. 5). The subset of data used for this analysis had no qualitative effect on the outcome of this analysis (Fig. S3). There was no evidence for spatial autocorrelation for any of the diversity metrics (Moran’s I, p > 0.05; Fig. 6).

Figure 5 Aggregate diversity metrics of each site plotted against distance from shore.

Both Simpson’s Index of evenness (A) and richness (B) are shown, and have been computed from the mean abundance of unique DNA sequences found across 4 PCR replicates at each of 24 sites. Lines and bands illustrate the fit and 95% confidence interval of a linear model.

Figure 6 Aggregate measures of diversity at each sample site.

Data are rarefied counts of mitochondrial 16S sequences collected from three parallel transects in Puget Sound, Washington, USA. Evenness (A) is the probability that two sequences drawn at random are different; richness (B) represents the total number of unique sequences from that location.

Taxon and life history patterns

We were able to assign a taxon name with confidence to 136 of 146 OTU sequences. The vast majority of sequences (97.6%) and OTUs (96.9%) were matched to organisms that have high potential for dispersal at either the gamete, larval, or adult stage, making it impossible to determine whether the source of that DNA was adults with well-documented spatial patterns (e.g., sessile nearshore specialists) or highly mobile early life history stages. Of the 6 OTUs for which dispersal is limited during all life history stages, only 2 occurred in more than two samples, precluding a quantitative comparison of spatial dispersion based on life history characteristics. These were assigned to Cymatogaster aggregata, a viviparous nearshore fish with internal fertilization, and Cupolaconcha meroclista, a sessile Vermetid gastropod with presumed internal fertilization and short larval dispersal (Strathmann & Strathmann, 2006; Phillips & Shima, 2010; Calvo & Templado, 2004). Cymatogaster aggregata was distinctly more abundant close to shore, with no sequences occurring in any sample beyond 250 m (Fig. 7). Cupolaconcha meroclista showed no such distinct spatial trend, occurring in nearly equal abundance at three sites, 75, 500, and 2,000 m from shore. An additional species that was highly abundant in the sequence data, the krill Thysanoessa raschii, has pelagic adults, highly seasonal reproduction, and sinking eggs; their distribution was consistent with our expectations based on a tendency of adults to aggregate offshore. Finally, the two most abundant taxa in the dataset were the mussel genus Mytilus and the Barnacle order Sessilia; the adults of both taxa are sessile and occur exclusively on hard intertidal substrata but have highly motile larvae. Because large-scale dispersal could not be ruled out for the vast majority of taxa, subsetting the community data by taxonomic group had no qualitative effect on the spatial patterning or diversity metrics, and we omit those results here.

Figure 7 Distribution of eDNA from select taxa.

Taxa represented are: Embiotocidae (A), Cupolaconcha meroclista (B), Thysanoessa raschii (C), Mytilus (D), Sessilia (E). Circles are colored and scaled by the proportion of that taxon’s maximum proportional abundance; the largest circle is the same size in each of the panels, and occurs where that taxon contributed the greatest proportional abundance of reads to that sample.

Discussion

Indirect surveys of organismal presence are a key development in ecosystem monitoring in the face of increased anthropogenic pressure and dwindling resources for ecological research. Monitoring of organisms using environmental DNA is an especially promising method, given the rapid pace of advancement in technological innovation and cost efficiency in the field of DNA sequencing and quantification. We document four key patterns: (1) eDNA communities far from one another tend to be less similar than those that are nearby, (2) distinct eDNA communities exist and are distributed in a non-random fashion, (3) diversity declines with distance from shore, and (4) spatial patterning of eDNA is associated with taxon-specific life history characteristics.

Communities far from one another tend to be less similar than those that are nearby

We demonstrate that distant locations have less-similar eDNA communities than proximate locations in Puget Sound, a dynamic marine environment. Our finding is in line with observations based on traditional surveys of terrestrial plants and fungi (Nekola & White, 1999; Bahram et al., 2013; Condit, 2002; Chust et al., 2006) and of microorganisms in freshwater (Wetzel et al., 2012), marine (Chust et al., 2013), and estuarine (Martiny et al., 2011) environments. To our knowledge, it is the first to report such a pattern using massively parallel sequencing of environmental DNA in the marine environment, and the first using any technique to describe this pattern from macrobial metazoans. We note that the theoretical expectation is that samples at very close distance be nearly completely similar, while our samples separated by the 50 m were only 40% similar. We interpret this to reflect the highly dynamic nature of this environment, which could cause DNA to be distributed quickly from its source, eroding the rise in similarity at small distances. At the same time, community similarity decreased to very low levels at larger scales, indicating that DNA distribution is not completely unpredictable. This finding implies that the effectively sampled area of individual water samples for eDNA analysis is likely to be quite small (<100 m) in this nearshore environment. Our estimated distance-decay relationship does indicate that proximate samples are more similar than distant samples, but we suggest this pattern is partially obscured by other factors, including signal from mobile, microscopic life-stages.

Distinct eDNA communities exist and are distributed in a non-random fashion

We demonstrate strong evidence for distinct community types and the non-random spatial patterning of those communities. While the spatial distributions of communities is surprising if one were concerned only with the macroscopic life stages of metazoans, it indeed does align with the broader view that even offshore pelagic communities are comprised of and influenced by nearshore organisms. This result underscores the idea that areas immediately offshore act as ecotones, a mixing zone of taxa characteristic of benthic and pelagic environments. While there was no distinct break in community types between onshore and offshore sites, there was some clustering of community types that may be explained by oceanographic features such as nearshore eddies generated by strong tidal exchange in a steep bathymetric setting (Yang & Khangaonkar, 2010). It would be useful to better understand such features during the period of sampling, by way of oceanographic monitoring devices. Finally, the uncertainty in identification of the number of distinct clusters to best characterize the community underlines the difficulty of identifying community patterns with the number of taxonomic groups considered here. We suspect that the signature of eDNA from microscopic life-stages may explain our inability to easily detect spatial community level patterns that align with our initial expectations.

Richness declines and evenness increases with distance from shore

We found that richness declined while evenness increased with distance from shore. Such a pattern is consistent with many other ecosystems which show strong clines in diversity metrics over environmental gradients. However, our study is novel in that it corroborates a cline well-known on macroscales for macrobiota on a much smaller spatial scale for microscopic animals, suggesting that there may be a self-similarity across scales in diversity patterning (Levin, 1992). The coastal ocean is a highly productive and diverse ecosystem, where biomass is concentrated most heavily along the bottom and shoreline (Ray, 1988). This differential in biomass concentration from the shoreline to open waters may contribute to the opposing trends we detected. Where particles (organisms, tissues, and cells) are sparse, fewer would be collected per sample of constant volume, thus decreasing the probability of drawing as many types (richness) and increasing the probability that any two particles originate from the same type (evenness). Intriguingly, the cline in diversity from inshore to offshore was not determined by shared changes in communities as one moved offshore; the classification analysis suggested a fair amount of differences among communities at a given offshore distance (Fig. 4).

Spatial patterning of eDNA is associated with taxon-specific life history

In contrast to our expectations, other taxa including species with sessile adult stages restricted to benthic hard substrates (e.g., barnacles, mussels) are among the most abundant taxa at sites furthest from shore. However, the larvae and gametes of these taxa are abundant, pelagic, and can be transported long distances by water movement (Strathmann, 1987). This indicates that we likely detected DNA of their pelagic phase gametes and larvae. It is always possible that DNA of adults was advected over long distances and detected offshore but in light of our results with krill and surfperch, we view this as unlikely. We interpret our results as evidence that the chaotic spatial distribution of eDNA communities (Fig. 4) results from our primers’ affinity for many species which at some point exist as microscopic pelagic gametes or larvae. Our results emphasize that expected results based on easily visually observed individuals or detectable with traditional sampling gear such as nets may be very different from results using eDNA. This does caution that eDNA surveys may have different purposes and may not be directly comparable to existing surveys (Shelton et al., 2016).

We acknowledge that sampling artifacts may have affected our results. For example if entire multicellular individuals were captured in our samples, their DNA could be in much greater density than eDNA, affecting the observed community. Our sampling bottles excluded particles larger than 500 micrometers, but gametes and very small larvae could have gained entry. It is possible that even a single small individual, containing many thousand mitochondria, would overwhelm the signal of another species from which hundreds of cells had been sloughed from many, larger individuals. Data on larval size distribution at the time of sampling from each species in our data set would allow us to estimate the frequency of such events. Nevertheless, it is precisely the sensitivity to small particles that makes the eDNA approach powerful, so we are reluctant to recommend that aquatic eDNA sampling use finer pre-filtering. Instead, we emphasize the importance of designing and selecting primer sets that selectively amplify target organisms. In the case of the present study, in order to recover patterns matching our expectations, this would be non-transient, benthic marine organisms lacking any pelagic life stage.

The marker we chose for this study detects a wide variety of metazoans while excluding other more common taxa; however, it does not effectively discriminate among species within a higher group in all cases. Other markers, such as mitochondrial cytochrome c oxidase subunit 1 (COX1, CO1, or COI) may provide adequate species-level resolution in some metazoan groups, but have other shortcomings including taxon dropout (Deagle et al., 2014) and amplification of more abundant non-metazoans, as we discovered in an accompanying study (Kelly et al., 2016a). Both have undesirable effects of biasing estimates of diversity. In our case, it is possible that the lumping of multiple species into one group underestimates the true richness of the group and of the entire sample, in turn obscuring true underlying patterns of diversity. In the case of COX1, well-documented primer biases cause failure to amplify some taxa, particularly in mixed samples, with the same result (Deagle et al., 2014). In fact, even surveys relying on traditional capture techniques (e.g., seine nets) and morphological characteristics are subject to biases imposed by the sampling gear (e.g., mesh size), the observer (e.g., taxonomic expertise), and organisms (e.g., morphologically cryptic species). Similarly, no single molecular marker adequately and effectively samples all taxa without bias (Drummond et al., 2015), and thus the choice of marker is an important and context-dependent one. Until whole-genome sequencing of individual cells is a reality, the tradeoffs between taxonomic breadth and resolution will continue to be problematic for metabarcoding studies, just as they are for more traditional ecological survey methods (Kelly et al., 2016a).

Our results also highlight the need for curated life-history databases. As technological advances increase the speed and throughput of DNA sequencing and sequence processing, making sense of these data in a timely manner requires that natural history data be stored in standard formats in centralized repositories. The rate at which we can make sense of high-throughput survey methods will be limited by our ability to collate auxiliary data. Databases such as Global Biodiversity Information Facility (GBIF), Encyclopedia of Life (EOL), and FishBase (Parr et al., 2014; Froese & Pauly, 2016) contain records of taxonomy, occurrence, and other rudimentary data types, but there is no centralized, standardized repository for even basic natural history data such as body size. As NCBI’s nucleotide and protein sequence database (GenBank) has facilitated transformative studies in diverse fields, an ecological analog would be a boon for biodiversity science.

Surveys based on eDNA are intensely scrutinized because of the danger that the final data are subject to complicated laboratory and bioinformatic procedures. Finding virtually no variability among lab and bioinformatic treatments from the point of PCR onward, we were confident our results represented actual field-based differences among samples. However, we note that one PCR replicate had a clear signal of contamination in that the sequence community was extremely similar to those from a different environmental sample. The source of this error is difficult to identify, but seems most likely to be an error during PCR preparation, either in assignment or pipetting during preparation of indexed primers. While the remainder of our results would be largely unchanged had we sequenced a single replicate per environmental sample, we believe the sequencing of PCR replicates is critical for ensuring data quality in eDNA sequencing studies.

While there is much work to be done before eDNA techniques can be confidently deployed as a standard method for ecological monitoring, this study serves as a first analysis of diversity at the fine spatial scales that are likely to be relevant to eDNA work in the field across a range of study systems.

Supplemental Information

Supplemental Information 1 Supplemental Material, including figures

Click here for additional data file.

We wish to thank Linda Park, Robert Morris, E. Virginia Armbrust, and James Kralj. The manuscript was improved by suggestions from editor Magnus Johnson, and reviewers Owen Wangensteen and Stephen Moss.

Additional Information and Declarations

Competing Interests

Author Contributions

DNA Deposition

Data Availability

The authors declare there are no competing interests.

James L. O’Donnell conceived and designed the experiments, performed the experiments, analyzed the data, wrote the paper, prepared figures and/or tables, reviewed drafts of the paper.

Ryan P. Kelly and Andrew Olaf Shelton conceived and designed the experiments, performed the experiments, reviewed drafts of the paper.

Jameal F. Samhouri, Natalie C. Lowell and Gregory D. Williams performed the experiments, reviewed drafts of the paper.

The following information was supplied regarding the deposition of DNA sequences:

All sequence data are available from NCBI under BioProject PRJNA338801.

The following information was supplied regarding data availability:

Analysis code and data (except sequence data) is available at: https://doi.org/10.5281/zenodo.242976.

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
