# Peer review of "Spatial distribution of environmental DNA in a nearshore marine habitat"

_PeerJ, doi:10.7717/peerj.3044_

## Round 0.1 · original submission · Minor Revisions

Both reviewers commented favourably on this manuscript and have made some useful and constructive comments which you should consider and respond to. I would also like you to respond to the comment by Bernd Haenfling on your pre-print as he is in agreement with reviewer #1 in relation to the degree of novelty this work represents (line 100 and your concluding sentence). Personally, I found your discussion of the results in the light of the advantages and limitations of this methodology interesting and highly informative. I wondered what the community analyses results would look like if you compartmentalised by e.g. life history or type (e.g. fish)?

·

Basic reporting

The article is well written. The structure is correct. The introduction and the material and methods sections are a little bit long, but can be easily improved by deleting some sentences (detailed below in this review). The references used are relevant and updated. The raw data and the analysis scripts are all publicly available.

Experimental design

The objectives are clearly established and the experimental design is perfect for addressing the proposed questions. The laboratory methods are rigorous and the analysis pipeline includes steps for assuring excellent quality of the resulting data set.

Validity of the findings

The findings are very robust. The high level of replication in the PCR analyses and the high-standards bioinformatic filtering steps used for removing any sequence attributable to possible contamination assure an excellent resulting data set to work with. The statistical analysis of the data are very clearly explained and the results are highly trustworthy.

Additional comments

The manuscript is, in general, very well written and the robust results obtained from the enhanced laboratory procedures and bioinformatic pipelines used are presented in a clear manner. I think that the introduction and material & methods sections are a little too long, but they can be enhanced by deleting some sentences (detailed below).

I think that it would be informative to add a paragraph in the discussion section commenting the caveats of using 16S as metabarcoding marker for taxonomic assignment of the MOTUs (i.e. the issues with the assignment of many sequences which can be assigned only to the family or lower levels), compared to other more variable markers with denser reference databases, such as COI, which, in most cases, allow assignments to the genus or species level.

Some specific corrections are detailed below:

L 36 to 40. This whole paragraph may be deleted to shorten the introduction, without any significant information loss.
L90. Correct “Actiniaria” and “Bryozoans”.
L91. Delete “(Decapoda)”. These habitats hold many other Crustacea other than Decapoda as well. Actually, I would write “barnacles (Sessilia) and other crustaceans” in line 89.
L93: “which provide”
L94: The microscopic plankton is not composed just by “diatoms and larvae”. Many important holoplanktonic organisms (e.g. copepods) could be mentioned here.
L100: I am not sure that this is “the first explicitly spatial analysis of eDNA-derived community similarity”, since some other papers have been published which reported patterns of spatial variability in marine systems (although using less quantitative approaches); e.g. De Vargas et al. 2015. I think that the authors should change this to “an explicitly spatial analysis of eDNA-derived community similarity”.
L117 to 122. This whole paragraph can be deleted. This methodology and the steps involved are now commonplace among aquatic biodiversity researchers and need not be enumerated.
L166 to 170. Why not use “µl” instead of “microliter”?
L169. I guess the authors mean “1.25 microliter genomic or eDNA template”.
L171 to 172. Use “95 ºC” instead of “95C”.
L195. “(machine NS500615, run 115, flowcell H3LFLAFXX)”. I don't think we need to know these details!
L196. “to enhance sequencing depth”. PhiX is added “to increase sequence diversity” and not to enhance sequencing depth. Actually, the more PhiX is spiked-in, the less amplicon reads will be obtained, thus decreasing (not enhancing) sequencing depth.
L223. Delete “a” in “This allows for a samples”.
L270-272. I think that these sentences may be confusing due to the use of some terms related to “dispersal range”, “type of fertilization” and “type of larval development”. Some “internally fertilized species” may have larvae with planktonic development and long dispersal capacity (e.g. some barnacles), whereas some broadcast spawners (e.g. some gorgonians) may be surface brooders with low dispersal ability. I think that the main traits affecting “dispersal range” is the type of larval development (planktonic or benthic) and the total planktonic larval duration (PLD), and not the type of fertilization or the strategy for gamete releasing, per se. I think that “internally fertilized species” in L271 could be changed to “benthic brooding species” and “broadcast spawners” could be changed to “species with long planktonic larval development”. Anyway, this semi-quantitative life-history information recorded by the authors in their data tables does not seem to have been used in their further analyses (since only qualitative considerations regarding two brooding species are discussed below).
L285. “333,537.9 ± 112,200.5”. It looks somehow awkward to use decimal values for numbers of reads here. You could round the values to “333,538 ± 112,200”
L314-315. You can delete “the probability that two reads chosen at random from a sample belong to different species”, since it has been already explained in the Materials & Methods section.
L344. “For the first time in a marine environment, we document four key patterns”. The authors should definitely use a more modest sentence here. At least the first two patterns they claim to report “for the first time” have been already reported previously in several works in the marine environment. E.g.: Fonseca et al. 2014, Guardiola et al. 2015, Lallias et a. 2015, De Vargas et al. 2015, etc., although possibly never with a sampling design specifically designed to get a quantitative assessment of distance decay.

·

Basic reporting

There were a couple of minor grammatical errors in the manuscript. 1) The final sentence (L292-294) of the “Sequence Data Processing (Bioinformatics)” paragraph on page 12 in the Results section. 2) The first sentence (L359) of the “Communities far from one another tend to be less similar than those that are nearby” paragraph on page 15 of the Discussion section.

I was a bit confused initially over discussion of massively parallel sequencing of 16S amplicons in relation to metazoan diversity in the abstract, and it wasn’t until into the introduction (L98) that I realised it was referring to 16S sequences from the mtDNA. Perhaps some clarity on this earlier on would be helpful? Perhaps also discussing the reasoning behind the decision to use 16S rDNA from the mtDNA rather than 18S rDNA would be helpful (e.g. Tang et. al., 2012; http://www.pnas.org/content/109/40/16208)?

I couldn’t find any reference to the details of the assigned clusters in Figure 3. I thought it might be useful to include a reference to figures or tables, perhaps as supplementary information, outlining the taxonomic composition of the clusters?

I felt the label for Figure 5 needed expanding to provide more detailed information on what it represents, making it more meaningful/understandable as a stand-alone figure.

The raw data is listed as available under Data Availability, though I assume this is still under embargo? The link on page 18 (L463) will need updating upon publication.

A GitHub DOI can be provided by Zenodo (https://zenodo.org) and will need updating upon publication.

Experimental design

When detailing how samples were collected (L126-128) under the Environmental Sampling header in the Methods section, I felt that information on how sampling artefacts were mediated (e.g. from L409-414) could have been provided sooner.

I felt a little more effort could have been made to facilitate the ability to replicate the analyses in this study. The methods section provided good information, but the analyses as detailed on L473 needs more clarity. An outline of the workflow steps and the input/output at each stage would be very helpful here.

Validity of the findings

In response to the “(3) Richness declines and evenness increases with distance from shore” heading in the Discussion section, I felt that decline in richness and increase in evenness might be expected due to dilution and osmotic factors in the ocean. Perhaps some reference to tidal flushing and dilution might be pertinent here?

I felt that under the remit of the Ecological Analyses headings throughout the study, there could have been some focus on reporting the alpha diversity and rarefaction analysis in relation to whether effective sequencing depth had been achieved for the samples in question.

Additional comments

I thought the paper was very interesting. The study was well designed, implemented, and written. It was useful to clarify the expectations and hypotheses outlined within, however, I felt more effort could have been made to provide recommendations on how some of the limitations in eDNA sampling could be overcome.

The bioinformatics/statistical methods seemed to be very well thought out, and consistent with current best practices in the analyses of this type of sequencing data.

---

## Round 0.2 · accepted · Accept

I am satisfied that you have dealt with the comments of the reviewers adequately. A group of us are hoping to instigate an e-DNA and marine management session at the 2018 ICES annual science conference. You may wish to present your work there?